# Functions of Hemp-Induced Exosomes against Periodontal Deterioration Caused by Fine Dust

**DOI:** 10.3390/ijms251910331

**Published:** 2024-09-25

**Authors:** Eunhee Kim, Yoonjin Park, Mihae Yun, Boyong Kim

**Affiliations:** 1Department of Food Science and Biotechnology, Andong National University, Andong 36729, Republic of Korea; 20225233@student.anu.ac.kr; 2Department of Bio-Hemp Technology, Andong Science College, Andong 36616, Republic of Korea; pyoonjin@naver.com; 3Department of Dental Hygiene, Andong Science College, Andong 36616, Republic of Korea; 4EVERBIO, 131, Jukhyeon-gil, Gwanghyewon-myeon, Jincheon-gun 27809, Republic of Korea

**Keywords:** cannabis, exosome, periodontal disease, PM10, periodontal ligament stem cell

## Abstract

Although fine dust is linked to numerous health issues, including cardiovascular, neurological, respiratory, and cancerous diseases, research on its effects on oral health remains limited. In this study, we investigated the protective effects of mature hemp stem extract-induced exosomes (MSEIEs) on periodontal cells exposed to fine dust. Using various methods, including microRNA profiling, PCR, flow cytometry, immunocytochemistry, ELISA, and Alizarin O staining, we found that MSE treatment upregulated key microRNAs, such as hsa-miR-122-5p, hsa-miR-1301-3p, and hsa-let-7e-5p, associated with vital biological functions. MSEIEs exhibited three primary protective functions: suppressing inflammatory genes while activating anti-inflammatory ones, promoting the differentiation of periodontal ligament stem cells (PDLSCs) into osteoblasts and other cells, and regulating LL-37 and MCP-1 expression. These findings suggest that MSEIEs have potential as functional biomaterials for applications in pharmaceuticals, cosmetics, and food industries.

## 1. Introduction

Fine dust, also known as particulate matter, is found in the air, and 2.5 μm (PM2.5) and 10 μm (PM10) particles are very harmful to humans and animals [1]. In particular, older adults, children, and patients with respiratory diseases are highly susceptible to fine dust [2]. Fine dust can cause various disorders, including cardiovascular, neurological, renal, reproductive, motor, systemic, respiratory, and cancerous diseases [3,4]. Pathogenic bacteria and viruses in PM10 enhance pathogenic infections in the oral environment [5,6,7]. Moreover, PM10 causes oral inflammation, cancer, and various other cancers in the human body [2]. Furthermore, gingival inflammation accelerates bacterial platelet aggregation, thereby elevating C-reactive protein and amyloid A fibrinogen levels in the liver as well as atherosclerosis in humans [8]. According to a recent study [9], fine dust suppresses the osteogenic differentiation of adipose-derived stem cells. Moreover, fine dust causes dermal tissue inflammation in humans and pets [9]. Skin inflammation results in the impairment of dermal immunity, leading to adipose-derived stem cell (ASCs) differentiation in the subcutaneous fatty tissues of the skin [9,10]. When exposed to fine dust, dermal cells upregulate apoptotic proteins, including BAX and CytC, and downregulate anti-apoptotic proteins, including AKT, P50, P52, and BCL-2 [9]. Inflammatory responses induced by fine dust hinder the maintenance of oral health in oral cells and, in particular, contribute to chronic oral diseases by impairing the functionality of stem cells [2].

Dental stem cells are classified into seven types—dental pulp stem cells, dental follicle progenitor cells, gingival mesenchymal stem cells, periodontal ligament stem cells (PDLSCs), stem cells for apical papilla, stem cells from human exfoliated deciduous teeth, and natal dental pulp stem cells [11]. PDLSCs can differentiate into various cell types, including cementoblasts, osteoblasts, fibroblasts, neurons, endothelial cells, cardiomyocytes, periodontal ligament cells (PDLCs), and pulp progenitor cells (PPCs) [12]. Periodontitis suppresses PDLSC differentiation into various types of cells [13]. The downregulation of cathelicidin (LL-37 peptide) in inflamed gingival cells enhances periodontium inflammation in neutrophils and various epithelial cells, including connective cells, fibroblasts, and gingival cells [14]. Recent reports have shown that cathelicidin suppresses the expression of monocyte chemoattractant protein-1 (MCP-1) in PDLCs under antimicrobial activity [15,16,17]. In contrast to MCP-1, cathelicidin activates the expression of secretory leukocyte protease inhibitors in PDLCs [18].

Exosomes of approximately 40–100 nm size are secreted from almost all cell types into the serum, urine, cerebrospinal fluid, ascites fluid, milk, and saliva [19,20] and contain various molecules, such as functional proteins, carbohydrates, mRNAs, microRNAs, and DNA [21]. In general, cells exposed to stimulants, such as phytoextracts, drugs, and pathogens [22], secrete exosomes with markedly altered components compared to those under unstimulated conditions [22]. Altered exosomes have adverse or beneficial effects on cells [22]. Furthermore, altered exosomes have various functions, including immune system modulation, prognostic biomarkers for diseases, and cancerous activity in the human body [20,22]. Based on these characteristics, induced exosomes (IEs) are considered biomaterials in various fields, including the pharmaceutical, cosmetic, and food industries [22].

*Cannabis sativa*, hemp, has various applications in foods, architectural materials, natural drugs, animal bedding and feed, textiles, clothing, bioplastics, and biofuel [23,24]. A recent study [25] reported an association between the pharmacological potential of cannabis and its antimicrobial activities, neuroprotection, management of gastrointestinal disorders, and anti-seizure and anti-cancer activities [25]. Mature hemp stem chemicals comprise fibers (23%), proteins (23%), ash (7%), CBD (0.001%), and THC (0.0064%); however, few reports have shown their biological activity [26,27]. Previously, phytochemical profiling of ethanol extracts [26,27] showed that tannin, saponin, and phenolic compounds (quercetin, apigenin, and rutin) were present in significantly high amounts in the seeds and stems. Among all the hemp components, the stem and roots are regarded as low-value pharmaceutical and cosmetic byproducts. Thus, further research is needed to demonstrate that the stem is a high-value product.

To date, no studies have examined both the functionality of exosomes derived from hemp stem extract and their protective effects on the oral cavity against fine dust exposure. This study aimed to investigate the bioactivities of mature hemp stem extract-induced exosomes (MSEIEs) derived from gingival cells exposed to mature hemp stem extract (MSE) against fine dust in gingival cells and PDLSCs.

## 2. Results

### 2.1. Protection of Gingival Cells against Fine Dust

Under MSE treatment, exosomes from gingival cells were isolated and purified using flow cytometry and immunofluorescence staining (Figure 1c). CC50 values for PM10 (16 µg/mL), MSE (1600 µg/mL), and MSEIEs (60 µg/mL) were determined (Figure 1). PM10 suppressed anti-apoptotic genes such as *AKT*, *NFκB-P50*, *NFκB-P52*, and *BCL2*, while MSE increased their expression. LC/MS/MS and ELISA analysis revealed concentrations of CBDVA, α-tocopherol, and flavonoids at 0.009 μg/mg, 0.624 μg/mL, and 12.37 μg/mL, respectively (Appendix A). MSE enhanced anti-apoptotic gene expression 3.8 times more than fine dust (Figure 2) and reduced apoptotic gene levels by 2.84 times. MSEIEs were even more effective, boosting anti-apoptotic gene expression 2.5 times more than MSE under fine dust exposure, and reducing apoptotic gene levels 8.8 times compared to MSE treatment (Figure 3).

### 2.2. Profiling of microRNAs in MSEIEs

The results of miRNA profiling (Figure 4 and Table 1) revealed that MSE altered the levels of various miRNAs associated with four categories, namely anti-inflammation, immune modulation, cellular differentiation, and homeostasis in MSEIEs. Compared with those in the exosomes induced under fine dust conditions, the hsa-miR-122-5p, hsa-miR-1301-3p, hsa-miR-196-3p, and hsa-let-7e-5p microRNAs were significantly upregulated in MSEIEs (Figure 4 and Table 1). Among these, hsa-miR-122-5P showed the most remarkable alteration in expression in the MSEIEs; its levels were approximately seven times higher than those of the FD-induced exosomes (Figure 4).

The radar charts display significantly altered levels of various miRNAs in exosomes derived from gingival cells. Con_IE, control-induced exosomes; FD_IE, fine dust-induced exosomes; MSEIEs, mature hemp stem extract-induced exosomes; MSE+FD_IE, induced exosomes from exposure to FD after MSE treatment (*p* < 0.05).

### 2.3. Activation of Osteogenic Differentiation by the MSEIEs

The two biomaterials activated PDLSC differentiation into osteoblasts, PDLCs, and PPCs despite exposure to fine dust (Figure 5 and Figure 6).

MSE-treated conditioned medium (MSECM) promoted PDLSC differentiation into osteoblasts, increasing osteogenesis by 3.9 times compared to fine dust exposure (Figure 5). Notably, MSEIEs treatment led to a 5.8-fold increase in osteogenesis despite fine dust exposure (Figure 6). MSEIEs also significantly enhanced PDLSC differentiation into PDLCs and osteoblasts, with increases of 1.6-fold and 2.4-fold, respectively, compared to MSECM (Figure 5 and Figure 6). Alizarin O staining confirmed these results, showing a 7.2-fold increase in osteocytic colony formation under MSEIEs treatment versus fine dust (Figure 7a), and a 1.4-fold increase compared to MSECM (Figure 7a). These findings were consistent with flow cytometry data (Figure 5 and Figure 6).

### 2.4. Activation of PDLC Differentiation by the MSEIEs

Among the three cell types, MSEIEs mostly activated the differentiation of PDLCs in fine dust. Compared to that in the control group, the differentiation induced by MSECM and MSEIEs was 1.8 and 4.8 times high, respectively (Figure 5 and Figure 6). Additionally, the number of differentiated PDLCs was 4.53 times higher in the MSECM-treated group than that under fine dust conditions (Figure 5). Surprisingly, the values under MSEIEs treatment were 3.53 times higher than those under the MSECM treatment (Figure 6). The results of immunocytochemistry (Figure 7b) were consistent with those of flow cytometry, wherein MSEIEs markedly activated PDLC differentiation despite exposure to fine dust (Figure 7b).

### 2.5. Activation of PPC Differentiation by the MSEIEs

Compared with the results for PDLCs and osteoblasts, although the differentiation to PPCs was the most attenuated under MSECM and MSEIEs treatments, MSEIEs significantly activated differentiation under fine dust conditions. The preventive effect was 2.6 times stronger than that under FD conditions (Figure 6).

### 2.6. Modulation of the Expression of Homeostatic Proteins by MSEIEs

Unlike the inactivated expression of MCP-1 in PDLSCs, MSE and MSEIEs activated the expression of the cathelicidin antimicrobial peptide (LL-37), a key protein that maintains the dental environment in gingival cells. The levels of LL-37 were 2.2 times higher in gingival cells under MSE *treatment* (Figure 8a), and MSECM attenuated the levels of MCP-1 by approximately 2.7 times compared with those under PM10 *conditions* (Figure 8b). Compared to MSE and MSECM *treatment*, MSEIEs *treatment* increased the expression of LL-37 by approximately 8.2 times in gingival cells and attenuated MCP-1 expression in PDLSCs by approximately 4.2 times (Figure 8a,b).

## 3. Discussion

In this study, two biomaterials derived from hemp were found to have biological functions against fine dust. The three main biological functions included anti-inflammatory activity, activation of differentiation, and modulation of the expression of proteins associated with maintenance of the dental environment.

Regarding their anti-inflammatory activity, MSE and MSEIEs suppressed the expression of inflammatory genes and activated that of anti-inflammatory genes in gingival cells despite the exposure to fine dust (Figure 2 and Figure 3). Compared to the MSE, MSEIEs strongly suppressed the expression of apoptotic genes while activating that of anti-apoptotic genes (Figure 2 and Figure 3). Flavonoids and tocopherols modulate the levels of miRNAs in cells [28,29]. Similar to that in these reports, the bioactive compounds in MSE (S2) have been suggested as candidate substances that play a key role in the upregulation of four miRNAs (hsa-miR-122-5p, hsa-miR-1301-3p, hsa-miR-196-3p, and hsa-let-7e-5p) in gingival cells. hsa-miR-122-5p is primarily expressed in the liver and has been extensively studied for its role in hepatic function and diseases. It is known to regulate cholesterol metabolism and hepatocellular carcinoma [30]. Recent studies have implicated hsa-miR-122-5p in the modulation of inflammatory responses. Specifically, hsa-miR-122-5p has been shown to target and suppress the expression of pro-inflammatory cytokines, such as IL-1β and TNF-α. By inhibiting the expression of these cytokines, hsa-miR-122-5p reduces inflammation, thereby exerting an anti-inflammatory effect [31]. Additionally, hsa-miR-122-5p can inhibit NF-κB signaling, a key pathway in the regulation of inflammatory responses, further contributing to its anti-inflammatory properties. Additionally, the upregulation of anti-inflammatory miRNAs in MSEIEs, shown in Figure 4 and Table 1, was consistent with the results shown in Figure 2 and Figure 3. Chronic inflammation caused by gingivitis leads to various disorders, including cardiovascular disease, respiratory diseases, diabetes, Alzheimer’s disease, cancer, multiple myeloma, and microbial carcinoma [32]. According to recent reports [33,34], every 5 μg/m^3^ increase in PM10 accelerates periodontitis occurrence and the presence of heavy metals in fine dust, causing degradation of the collagen matrix, inhibition of mineralization, and calcium resorption in the dental environment. In addition to outdoor fine dust pollution, indoor air pollution also threatens the health of humans and pets [35,36]. Our results (Figure 2 and Figure 3) suggest that these two biomaterials are suitable for preventing periodontitis.

Furthermore, the two materials protected and activated the differentiation of PDLSCs despite their exposure to fine dust. These materials activated the differentiation of PDLSCs to osteoblasts, PDLCs, and PPCs (Figure 5, Figure 6 and Figure 7). Notably, MSEIE was more effective than MSECM for osteogenic and PDLC differentiation (Figure 5 and Figure 6). These comparative results demonstrated the potential of MSEIE as a bio-pharmaceutical material. According to previous reports [37,38,39,40], four upregulated miRNAs (Figure 4), including hsa-miR-122-5p, hsa-miR-1301-3p, hsa-miR-196-3p, and hsa-let-7e-5p, promote osteogenic and other types of cellular differentiation. Furthermore, various studies on functional miRNAs [41,42] have shown that hsa-miR-122-5p plays a predominant role in liver function and metabolism; however, recent research has uncovered its involvement in osteogenic differentiation. This miRNA modulates the expression of genes critical for osteoblast differentiation. Specifically, hsa-miR-122-5p targeted runt-related transcription factor 2 (RUNX2), a master regulator of osteogenesis. hsa-miR-1301-3p is a positive regulator of osteogenic differentiation. This miRNA enhances the osteogenic potential of MSCs by targeting specific osteogenic inhibitors. One such target is SMAD6, an inhibitory SMAD protein that negatively regulates bone morphogenetic protein (BMP) signaling. By suppressing *SMAD6* expression, hsa-miR-1301-3p promoted BMP signaling, leading to enhanced osteoblast differentiation and mineralization. hsa-miR-196-3p plays a significant role in the modulation of osteogenic pathways through the regulation of *HOX* gene clusters, which are crucial for skeletal development. This miRNA targets specific *HOX* genes that are involved in the differentiation of osteoprogenitor cells. By fine-tuning the expression of these genes, hsa-miR-196-3p influenced the commitment of MSCs to the osteogenic lineage. Moreover, hsa-miR-196-3p has been shown to interact with the BMP and Wnt signaling pathways, both of which are vital for osteoblast differentiation [43]. The let-7 family, including hsa-let-7e-5p, is known to play a role in maintaining cellular homeostasis and differentiation. hsa-let-7e-5p specifically regulates osteogenic differentiation by targeting and downregulating *HMGA2*, a gene that negatively affects osteoblast differentiation [44].

PDLSCs are promising cells for periodontal regeneration, osteogenesis, neural damage repair, and periodontal therapy [45,46]. They are key players in periodontal regeneration therapies, which use these cells for safe and fast recovery after treatment [47]. In implant therapy, suppression of peri-implantitis is crucial for achieving a high therapeutic success rate [48]; bone loss due to inflammation increases the rate of failure. Additionally, osseointegration through the activation of osteogenesis enhances the anchoring and integration of implant fixtures [49]. Therefore, the reported bioactivities of MSE and MSEIEs are crucial for preventing periodontitis and increasing the implant success rates. PPCs differentiate into various types of cells, including adipocytes, myocytes, osteocytes, chondrocytes, odontoblast-like cells, and neural cells [50]. Our results (Figure 5 and Figure 6) showed that the differentiation of PDLSCs into PPCs was most highly attenuated under treatment with the two materials. This may be attributed to the fact that MSEIEs and MSE activate the differentiation to PDLCs and osteoblasts. Dental pulp plays various roles, including blood supply, maintenance of hemostasis under various stimuli, dentin formation, and providing nourishment and moisture to teeth [51]. The functions of the two materials help maintain a healthy pulp and protect the pulp against fine dust.

Third, the two materials enhanced the expression of modulating proteins in healthy dental environments. Despite exposure to fine dust, MSEIEs and MSE activated LL-37 upregulation and MCP-1 downregulation in gingival cells and PDLSCs, respectively (Figure 8a,b). The functional analysis of the four miRNAs in MSEIEs (Figure 4) suggested that these miRNAs play a crucial role as homeostatic modulators in gingival cells and PDLSCs exposed to fine dust. Although the MSE and MSECM significantly suppressed MCP-1 protein expression in PDLSCs, LL-37 was markedly upregulated upon MSEIE treatment (Figure 8a). The LL-37 peptide has various functions, including oral microbiota control, pro-apoptotic activity, immunomodulation, antimicrobial activity, and promotion of wound healing [52]. This protein maintains homeostasis in the oral cavity through its biological functions. Compared with that under healthy conditions, the concentration of LL37 is two times high in the blood during the pro-apoptotic state and eight times higher during the anti-proliferative state in chronic apical periodontitis [53]. MSE and MSEIEs treatments increased the concentrations of LL-37 by approximately 4 and 10 times, respectively, compared with those in the controls (Figure 8a). These results suggest that MSE and MSEIEs control pro-apoptotic and anti-proliferative states, respectively, in addition to maintaining homeostasis in the oral cavity. hsa-miR-1301-3p is a regulator of immune cell activation. This miRNA is involved in the modulation of T-cell- and B-cell responses. Studies have demonstrated that hsa-miR-1301-3p targets specific molecules involved in immune activation such as CD40 and CD80. By regulating the expression of these co-stimulatory molecules, hsa-miR-1301-3p influences the activation and proliferation of immune cells. Additionally, hsa-miR-1301-3p affects the differentiation of naïve T cells into effector T cells, thereby playing a crucial role in shaping the adaptive immune response [54]. MCP-1 is involved in various diseases, including Alzheimer’s disease, Parkinson’s disease, multiple sclerosis, diabetes, tuberculosis, COVID-19, and rheumatoid arthritis [55]. In the oral environment, LL37 secreted from gingival cells suppresses the expression of *MCP-1* in PDLCs [56]. Remarkably, compared with that under PM10 conditions, MSEIEs treatment markedly suppressed the expression of *MCP-1* (Figure 8b). The let-7 family of miRNAs, including hsa-let-7e-5p, is known to play a role in maintaining cellular homeostasis and regulating cell cycle progression. hsa-let-7e-5p has been shown to target several genes involved in cell proliferation and apoptosis, such as *RAS* and *MYC* [57]. By controlling the expression of these genes, hsa-let-7e-5p ensures a balance between cell growth and death, thus maintaining homeostasis. Additionally, hsa-let-7e-5p has been implicated in the regulation of immune homeostasis by modulating the expression of cytokines and growth factors that are critical for immune cell function and survival [58]. Overall, our results suggest that the two materials maintain a homeostatic environment by modulating the expression of these two proteins in the oral cavity.

## 4. Materials and Methods

### 4.1. Cell Culture and Cytotoxicity Test to Establish the Treatment Dosage

Normal human gingival cells (PCS-201-018, ATCC, VA, USA) were cultured in complete growth media (PCS-201-030 and PCS-201-041, ATCC) at 37 °C and 5% CO_2_. The cultured cells were treated with MSE and fine dust ERM-CZ100 (ERM, Sigma-Aldrich, St. Louis, MO, USA), containing 17 types of polycyclic aromatic hydrocarbons (PAHs), for one day at 37 °C and 5% CO_2_. The MSE was prepared via evaporation from 50% ethanol and steam extraction (4 h at 70 °C) of mature hemp stems from Cheongsam cultured in Andong City, Korea. The extracts were concentrated and sterilized using a rotary evaporator (DAIHAN Scientific, Seoul, Republic of Korea) and a 0.2 μm syringe filter (Nalgene, Waltham, MA, USA). MSEIEs were isolated from gingival cells at an established concentration (1600 µg/mL). To avoid contamination from foreign extracellular vesicles, PDLSCs (SKU: 36085-01 and M36085-01S, Celprogen, Torrance, CA, USA) were cultured without fetal bovine serum (FBS). Exosomes were isolated and purified from the supernatants using an exoEasy Maxi Kit (QIAGEN, Hilden, Germany) and a CD68 Exo-Flow Capture Kit (System Biosciences, Palo Alto, CA, USA). Based on the recommendations of the International Society for Extracellular Vesicles, the extent of purification was evaluated using marker expression (flow cytometry) and imaging (immunostaining). The cultured PDLSCs were exposed to MSEIEs for one day to determine the treatment dosage. The exposed cells were stained using the Annexin V-PI apoptosis detection kit (Invitrogen, Carlsbad, MA, USA), and cytotoxicity was evaluated using a flow cytometer (BD FACScalibur, BD Biosciences, San Jose, CA, USA) and FlowJo 10.6.1 (BD Biosciences) to establish the cytotoxic concentration (CC_50_).

### 4.2. Anti-Apoptotic Activity of MSE

Total RNA was isolated from gingival cells cultured under various conditions (Con, MSE, PM10, and MSE+PM 10) using the RiboEx reagent (GeneAll, Seoul, Republic of Korea), and cDNA was synthesized from the isolated RNA using Maxime RT PreMix (iNtRON, Seongnam, Republic of Korea). cDNA was amplified with primers (Table 2) using conventional PCR (PCR PreMix, Bioneer, Daejeon, Republic of Korea) under the following cycling conditions: 1 min at 95 °C, followed by 35 cycles of 35 s at 59 °C and 1 min at 72 °C. The expression levels of amplified target genes in the samples were determined by normalizing to those of the housekeeping gene *glyceraldehyde-3-phosphate dehydrogenase* (*GAPDH*). The amplified DNA was estimated using iBright FL1000 and iBright Analysis Software 4.0.0 (Invitrogen).

### 4.3. Profiling of microRNA in MSEIEs

The isolated MSEIEs were sequenced by Ebiogen Inc. (Seoul, Republic of Korea) to analyze exosomal function. An Agilent 2100 bio-analyzer and RNA 6000PicoChip (Agilent Technologies, Amstelveen, The Netherlands) were used to evaluate RNA quality. The RNA was quantified using a NanoDrop 2000 spectrophotometer (Thermo FisherScientific, Waltham, MA, USA). Small RNA libraries were prepared and sequenced using an Agilent 2100 Bio-analyzer instrument for the high-sensitivity DNA assay (Agilent Technologies, Inc., Santa Clara, CA, USA) and the NextSeq500system single-end 75 sequencing (Illumina, San Diego, CA, USA). To obtain an alignment file, the sequences were mapped using Bowtie 2 software (CGE Risk, Lange Vijverberg, The Netherlands), and the read counts were extracted from the alignment file using bedtools (v2.25.0) (GitHub, Inc., San Francisco, CA, USA) and the R language (version 3.2.2) (R Studio, Boston, MA, USA) to evaluate miRNA expression levels. miRWalk 2.0 (Ruprecht-Karls-Universität Heidelberg, Medizinische Fakultät Mannheim, Germany) was used for miRNA target signal analysis, and ExDEGA v.2.0 (ebiogen Inc., Seoul, Republic of Korea) was used to deduce radar charts.

### 4.4. PDLSC Differentiation Patterns under MSECM Treatment

After culturing the PDLSCs (Celprogen, Torrance, CA, USA) in specific media (SKU: M36085-01S) under different conditions (Con, PM10, MSECM, or MSECM+PM10) for a day, the cultured cells were fixed in 2% paraformaldehyde for 4 h and treated with 0.02% Tween 20 for 5 min. The treated cells were incubated with three fluorescence-conjugated immunoglobulins—FITC-anti-asporin (Abbexa, Cambridge, UK), PE-anti-osteopontin (R&D Systems, Minneapolis, MN, USA), and APC-anti-cytokeratin 4 (Biorbyt, Cambridge, UK)—at 37 °C for 2 days. The stained cells were evaluated using a flow cytometer (BD FACScalibur), FlowJo 10.6.1 (BD Biosciences), and Prism 7 (GraphPad, San Diego, CA, USA).

### 4.5. Immunocytochemistry for Osteoblasts

After culturing the PDLSCs under different conditions (PM10, MSECM, MSECM+PM10, MSEIEs, or MSEIEs+PM10), the cultured cells were fixed with 2% paraformaldehyde for 12 h and stained using Alizarin O reagent (Sigma, St. Louis, MO, USA) for 40 min. The stained cells were evaluated using a fluorescence microscope (Eclipse Ts-2, Nikon, Shinagawa, Japan) and the imaging software NIS-elements V5.11 (Nikon)

### 4.6. Localization of PDLSC Markers Using Immunocytochemistry

After culturing the PDLSCs under different conditions (Con, PM10, MSECM, MSECM +PM10, MSEIEs, or MSEIEs+PM10), the cultured cells were fixed in 2% paraformaldehyde for 12 h and treated with 0.02% Tween 20 for 10 min. The treated cells were incubated with three fluorescence-conjugated immunoglobulins and FITC-anti-asporin (Abbexa, Cambridge, UK). The stained cells were evaluated using a fluorescence microscope (Eclipse Ts-2, Nikon, Shinagawa, Japan) and the imaging software NIS-elements V5.11 (Nikon)

### 4.7. Statistical Analysis 

To ensure the reliability of the data, we conducted three independent reproducibility experiments, with five repeated measurements for each experiment. All data were analyzed using a one-way analysis of variance (ANOVA) with post hoc (Scheffe’s method) using Prism 7 software (GraphPad, San Diego, CA, USA).

## 5. Conclusions

This study highlights the substantial potential of a hemp-derived biomaterial, specifically MSEIEs, to counteract the detrimental effects of fine dust on dental health. This investigation revealed three primary biological functions of MSEIEs—anti-inflammatory activity, activation of differentiation, and modulation of the expression of proteins associated with maintaining a healthy dental environment. MSEIEs demonstrated robust anti-inflammatory properties by suppressing the expression of inflammatory genes and activating that of anti-inflammatory genes in gingival cells exposed to fine dust. Furthermore, MSEIEs promoted and protected the differentiation of PDLSCs into osteoblasts, PDLCs, and PPCs despite exposure to fine dust. Additionally, MSEIEs enhanced the expression of modulating proteins, such as LL-37 and MCP-1, which are crucial for maintaining oral homeostasis.

Although MSEIEs demonstrated excellent functionality at the cellular level, results from animal studies and clinical trials are necessary for its clinical application. Future research should focus on clinical trials to validate the in vitro findings of the present study. Investigating the long-term effects of MSEIEs in real-world dental applications, including their efficacy in preventing periodontitis and enhancing the implant success rate, is crucial. Additionally, exploring the molecular mechanisms underlying the modulation of the expression of miRNAs and proteins by MSEIEs will provide a deeper insight into their therapeutic potential. The findings of this study offer promising prospects for the development of novel dental treatments that utilize hemp-derived biomaterials. MSEIEs can potentially revolutionize the management of dental disorders caused by environmental pollutants, particularly fine dust. Owing to their effective anti-inflammatory and differentiation-promoting properties, MSEIEs can enhance periodontal health, improve the outcomes of dental implants, and contribute to the overall well-being of individuals exposed to fine dust. From an industrial perspective, MSEIEs has the potential to be utilized as a functional material in oral care products and implant-related products for humans, as well as in oral care products for pets. This study paves the way for innovative, natural, and effective solutions for dental care.

## Figures and Tables

**Figure 1 ijms-25-10331-f001:**
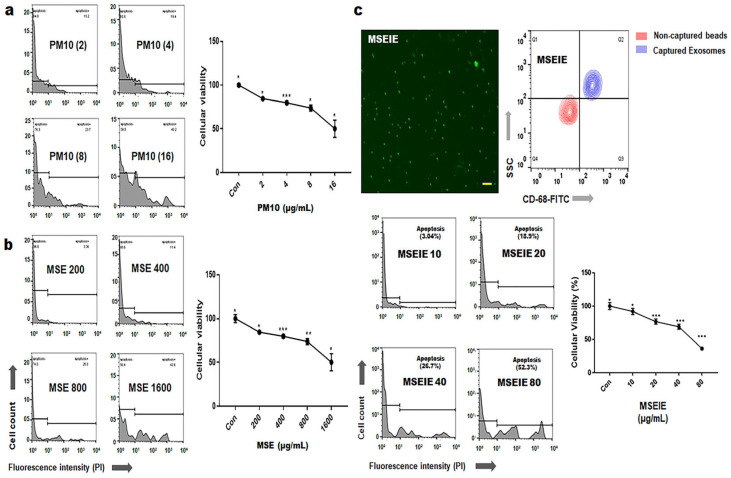
Treatment dosages of fine dust, mature stem extract (MSE), and MSE-induced exosomes (MSEIEs). (**a**,**b**) Cytotoxic concentration (CC_50_) of fine dust (PM10) and MSE in gingival cells. (**c**) MSEIEs treatment dose in periodontal ligament stem cells (PDLSCs). Con; control, (* *p* < 0.05; ** *p* < 0.01; *** *p* < 0.001) (scale bar = 20 μm).

**Figure 2 ijms-25-10331-f002:**
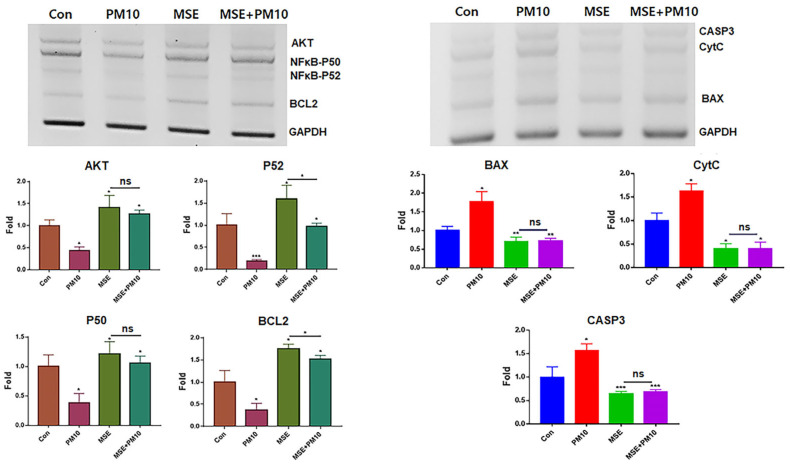
Levels of anti-apoptotic and apoptotic genes in gingival cells under MSE treatment. Levels of apoptotic (*BAX*, *CytC*, and *CASP3*) and anti-apoptotic (*AKT*, *NFκB-P50*, *NFκB-P52*, and *BCL2*) genes in gingival cells under MSE and fine dust treatment. MSE+PM10, PM10 treatment after MSE exposure, ns; not significant (* *p* < 0.05, ** *p* < 0.01, *** *p* < 0.001).

**Figure 3 ijms-25-10331-f003:**
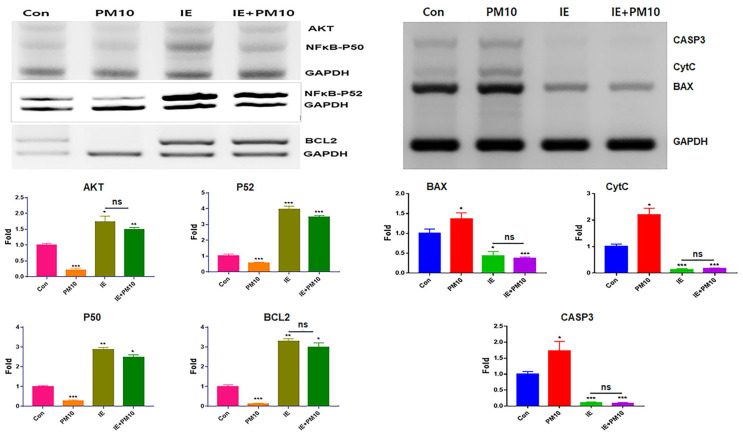
Levels of anti-apoptotic and apoptotic genes in gingival cells after MSEIEs treatment. Levels of apoptotic (*BAX*, *CytC*, and *CASP3*) and anti-apoptotic (*AKT*, *NFκB-P50*, *NFκB-P52*, and *BCL2*) genes in gingival cells treated with MSEIEs, ns; not significant (* *p* < 0.05, ** *p* < 0.01, *** *p* < 0.001).

**Figure 4 ijms-25-10331-f004:**
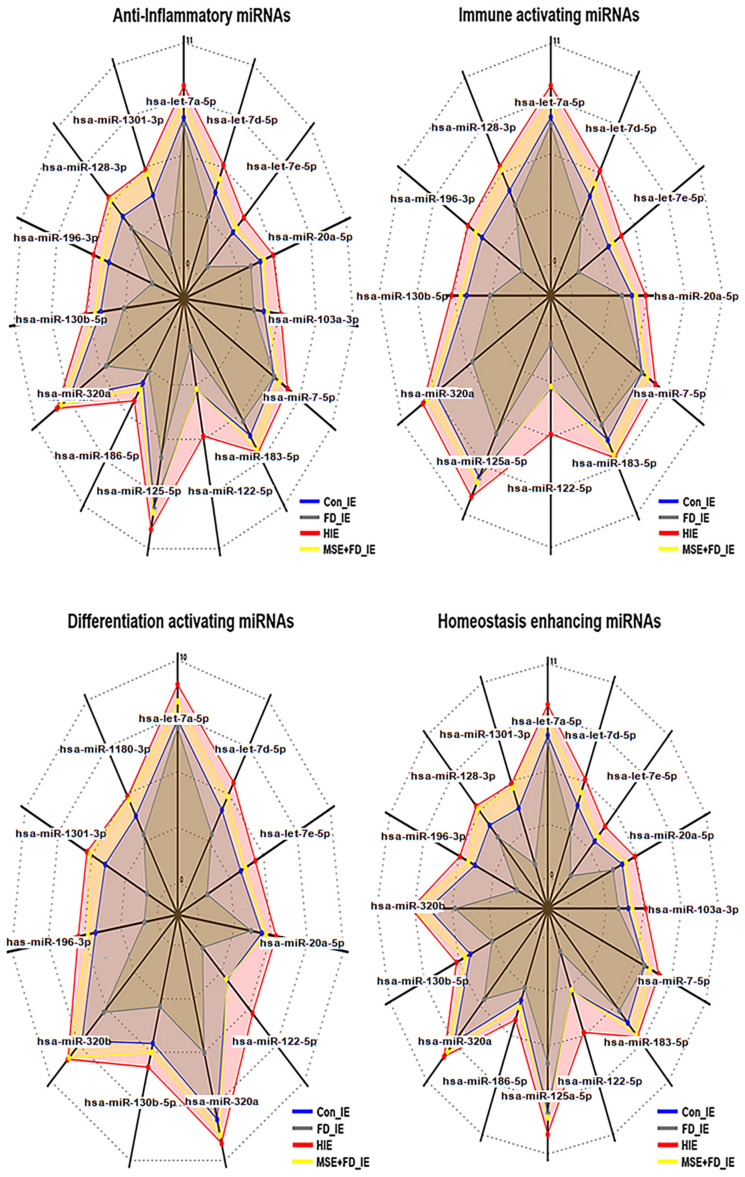
Profiling of miRNAs in various exosomes.

**Figure 5 ijms-25-10331-f005:**
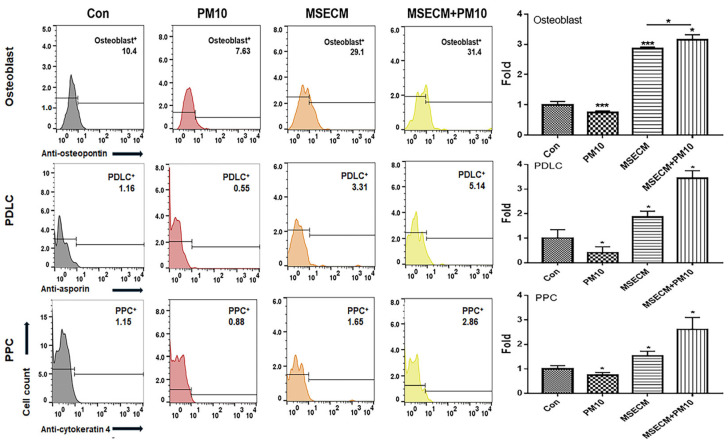
Differentiating patterns of PDLSCs cultured in an MSE-conditioned medium. Differentiation of PDLSCs into osteoblasts, periodontal ligament cells (PDLCs), and pulp progenitor cells (PPC). MSECM, matured hemp stem extract-conditioned medium; PM10, fine dust (* *p* < 0.05, *** *p* < 0.001).

**Figure 6 ijms-25-10331-f006:**
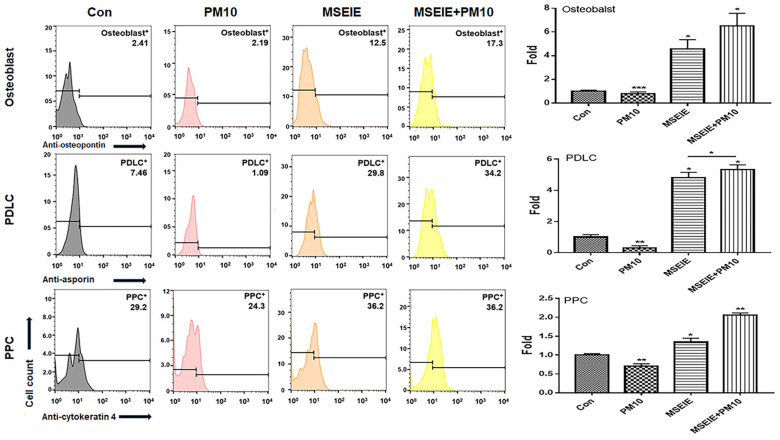
Differentiation patterns of PDLSCs under MSECM and MSEIEs treatments. Differentiation of PDLSCs into osteoblasts, PDLCs, and PPC MSEIEs from gingival cells under MSE, MSEIEs+PM10, and PM10 treatments after exposure to MSEIEs (* *p* < 0.05, ** *p* < 0.01, *** *p* < 0.001).

**Figure 7 ijms-25-10331-f007:**
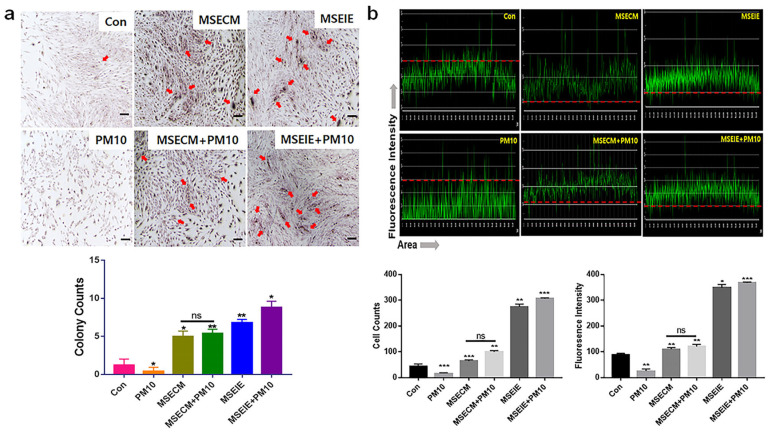
Immunocytochemistry results for differentiation to osteoblasts or PDLCs from PDLSCs upon treatment with the two biomaterials. (**a**) Images of osteoblast cells and colony formation (red arrows) assessed using alizarin staining. (**b**) Immunocytochemical results with the anti-asporin PDLC marker conjugated with green fluorescence. The dashed red lines show the fluorescence intensity (median value) of the control. The stained cells and colonies were counted using the NIS-elements V5.11 software (ns; not significant, * *p* < 0.05, ** *p* < 0.01, *** *p* < 0.001) (scale bar = 20 μm).

**Figure 8 ijms-25-10331-f008:**
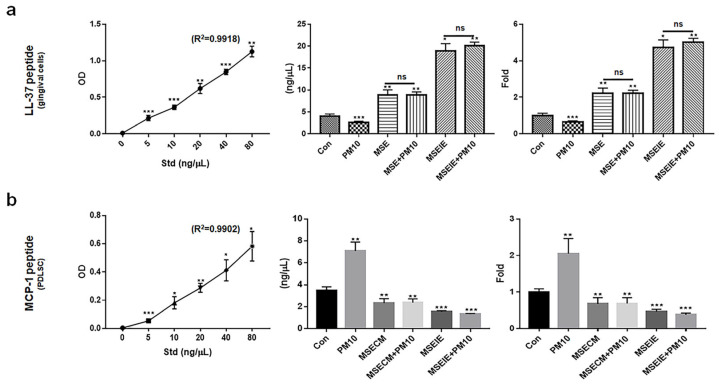
Expression of homeostatic proteins under various conditions. (**a**) Levels of LL-37 peptide in gingival cells under various conditions. (**b**) Levels of MCP-1 protein in PDLSCs under various conditions (ns; not significant, * *p* < 0.05, ** *p* < 0.01, *** *p* < 0.001).

**Table 1 ijms-25-10331-t001:** Profiling of significant microRNAs in MSEIEs.

Category	Anti-Inflammation	Immnue Activation	Differentiating Activation	Homeostatic Enhancement
Up-regulated genes	16	11	11	15
Dramatic Up-regulation	hsa-miR-1301-3p	hsa-miR-196-3p	hsa-miR-1301-3p	hsa-miR-1301-3p
	hsa-miR-196-3p	hsa-let-7e-5p	hsa-miR-196-3p	hsa-miR-196-3p
	hsa-let-7e-5p	hsa-miR-122-5p	hsa-let-7e-5p	hsa-let-7e-5p
	hsa-miR-122-5p		hsa-miR-122-5p	hsa-miR-122-5p

**Table 2 ijms-25-10331-t002:** Sequences for PCR primers.

Gene	Seq (5′ → 3′)
*AKT*	F: GGCTGCCAAGTGTCAAATCC R: AGTGCTCCCCCACTTACTTG
*NFκB-P50*	F: CGGAGCCCTCTTTCACAGTT R: TTCAGCTTAGGAGCGAAGGC
*NFκB-P52*	F: AGGTGCTGTAGCGGGATTTC R: AGAGGCACTGTATAGGGCAG
*BCL2*	F: CTGCTGACATGCTTGGAAAA R: ATTGGGCTACCCCAGCAATG
*BAX*	F: AGCGCTCCCCCACTTACTTG R: GACAGGGACATCAGTCGCTT
*CytC*	F: ATGAATGACCACTCTAGCCA R: ATAGAAACAGCCAGGACCGC
*CASP3*	F:TCCCTGGGAAGAAAGAGTTGTGG R:TGAACATGGCACCTCTGCAAC
*GAPDH*	F: GTGGTCTCCTCTGACTTCAACA R: CTCTTCCTCTTGTGCTCTTGCT

## Data Availability

Data are contained within the article and Appendix A.

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
