# Peer review of "Functions of Hemp-Induced Exosomes against Periodontal Deterioration Caused by Fine Dust"

_ijms, 2024, doi:10.3390/ijms251910331_

Round 1

Reviewer 1 Report

Comments and Suggestions for Authors

The study reports on the effects of hemp stem extract-induced exosomes on periodontal cells exposed to fine dust. The novelty of this research is underscored by the fact that no similar articles have been indexed in PubMed. Despite of its novelty the authors should highlight why this paper should be relevant for the readers.

The manuscript is generally well written, and the use of various techniques, including miRNA profiling, flow cytometry, and ELISA, strengthens the validity of the findings. However, several areas of improvements could enhance the article impact and readability.

The abstract provides a comprehensive overview of the study's objectives and key findings. However, it could benefit from conciseness, particularly in reducing the number of technical details which could overwhelm readers unfamiliar with the field. 

The introduction effectively introduces the research problem, but some transitions between topics (i.e., from environmental impacts of fine dust to dental stem cells) could be smoother. Please report on how your contribution fills the existing research gaps. Additionally, citing more recent literature (from 2023–2024) would further highlight the timeliness of the research.

The methodology is well-detailed, allowing for reproducibility and follow the guidelines proposed for cell studies. A justification for the sample size or discussing statistical power should be introduced. 

In the results section most of the figures, could benefit from larger font sizes or clearer annotations to improve readability. There is also a degree of repetition in the description of gene expression changes, which could be condensed. 

In the discussion section the authors should address the limitations of the study more explicitly. For instance, the potential variability in MSEIE efficacy due to different levels of fine dust exposure could be discussed. Please consider to report on the potential commercial and clinical applications of MSEIEs, as well as the challenges in translating these findings to practice. The authors could also consider to suggest specific directions for future research, such as in vivo studies or clinical trials, to validate these in vitro findings.

Please ensure consistent use of terminology throughout the text, particularly regarding the descriptions of cellular processes and gene expression changes.

Comments on the Quality of English Language

Minor changes are needed

Author Response

Thank you for your excellent comments, which have been very helpful in improving our manuscript. In the revised version, the changes have been underlined in red and revised parts for common comments by reviewers have been underlined in yellow.

Comments1] The study reports on the effects of hemp stem extract-induced exosomes on periodontal cells exposed to fine dust. The novelty of this research is underscored by the fact that no similar articles have been indexed in PubMed. Despite of its novelty the authors should highlight why this paper should be relevant for the readers.

Answer1] We described the goals and highlight for our study at introduction and conclusion sections

Comments2] The abstract provides a comprehensive overview of the study's objectives and key findings. However, it could benefit from conciseness, particularly in reducing the number of technical details which could overwhelm readers unfamiliar with the field.

Answer2] We have simplified and condensed the abstract to make it easier to understand.

Comments3] The introduction effectively introduces the research problem, but some transitions between topics (i.e., from environmental impacts of fine dust to dental stem cells) could be smoother. Please report on how your contribution fills the existing research gaps. Additionally, citing more recent literature (from 2023–2024) would further highlight the timeliness of the research.

Answer3] We revised the introduction based on your comments and the sections where it was possible to apply more recent references have been updated with the latest literature

Comments4] The methodology is well-detailed, allowing for reproducibility and follow the guidelines proposed for cell studies. A justification for the sample size or discussing statistical power should be introduced.

Answer4] We described the statistical power at the materials and methods section.

Comments5] In the results section most of the figures, could benefit from larger font sizes or clearer annotations to improve readability. There is also a degree of repetition in the description of gene expression changes, which could be condensed.

Answer5] We have revised the figures that needed improved readability (Figures 1, 2, 3, 5, and 8), and made the repetitive explanations more concise.

Comments6] In the discussion section the authors should address the limitations of the study more explicitly. For instance, the potential variability in MSEIE efficacy due to different levels of fine dust exposure could be discussed. Please consider to report on the potential commercial and clinical applications of MSEIEs, as well as the challenges in translating these findings to practice. The authors could also consider to suggest specific directions for future research, such as in vivo studies or clinical trials, to validate these in vitro findings. Please ensure consistent use of terminology throughout the text, particularly regarding the descriptions of cellular processes and gene expression changes.

Answer6] We added the description based on your comments at the conclusion section and double checked and revised the terminology without the underlines

Reviewer 2 Report

Comments and Suggestions for Authors

Dear authors,

I enjoyed reading your article. The topic is very interesting, the way you treated the subject as well. I also find the results you obtained interesting. I appreciate the graphic quality of the article. I also congratulate you on the choice of the bibliography which is up-to-date and relevant for the chosen topic. Also, the desire to continue in vivo research is welcome.

However, I would like you to specify a few aspects:

-I did not find the limits of the study, maybe try to specify this aspect as well

-Have you also found adverse effects/contraindications to the use of MSEIEs?

Author Response

Thank you for your excellent comments, which have been very helpful in improving our manuscript. In the revised version, the changes have been underlined in orange and revised parts for common comments by reviewers have been underlined in yellow.

Comments1] I did not find the limits of the study, maybe try to specify this aspect as well

Answer1] We added the limits of the study at the conclusion section in this manuscript

Comments2] Have you also found adverse effects/contraindications to the use of MSEIEs?

Answer2] Hemp stem or root extracts can be used as ingredients in cosmetics, and our experiments have shown that their toxicity is significantly low, indicating high skin safety. However, in countries where the regulation of narcotics is strict, there are limitations to using hemp stem extracts as functional biomaterials related to the human oral cavity. Nevertheless, for pets, these regulations are less restrictive, allowing for potential applications in this area first. Furthermore, while hemp-derived exosomes may raise issues related to narcotic regulations, exosomes induced from human-derived cells stimulated by hemp are free from these regulatory constraints. Additionally, exosomes can be developed into bio-pharmaceutical materials.